# Resveratrol and Some of Its Derivatives as Promising Prophylactic Treatments for Neonatal Hypoxia-Ischemia

**DOI:** 10.3390/nu14183793

**Published:** 2022-09-14

**Authors:** Hélène Roumes, Pierre Goudeneche, Luc Pellerin, Anne-Karine Bouzier-Sore

**Affiliations:** 1Centre de Résonance Magnétique des Sysytèmes Biologiques (CRMSB), UMR 5536, University of Bordeaux and CNRS, F-33000 Bordeaux, France; 2Ischémie Reperfusion, Métabolisme et Inflammation Stérile en Transplantation (IRMETIST), Inserm U1313, University of Poitiers and CHU Poitiers, F-86021 Poitiers, France

**Keywords:** polyphenols, neonatal hypoxia-ischemia, neuroprotection

## Abstract

Due to the rate of occurrence of neonatal hypoxia-ischemia, its neuronal sequelae, and the lack of effective therapies, the development of new neuroprotective strategies is required. Polyphenols (including resveratrol) are molecules whose anti-apoptotic, anti-inflammatory, and anti-oxidative properties could be effective against the damage induced by neonatal hypoxia-ischemia. In this review article, very recent data concerning the neuroprotective role of polyphenols and the mechanisms at play are detailed, including a boost in brain energy metabolism. The results obtained with innovative approaches, such as maternal supplementation at nutritional doses, suggest that polyphenols could be a promising prophylactic treatment for neonatal hypoxia-ischemia.

## 1. Introduction

Neonatal hypoxia-ischemia (HI) is a major public health problem, both in terms of its occurrence (in 1% to 6% of births in developed countries) and its subsequent adverse effects, such as lethality, cognitive disorders, and/or motor disabilities. It usually results from a prolapse of the umbilical cord, a uterine rupture, or maternal hypotension. The disruption of a newborn’s cerebral blood flow is the primary cause of brain injury. The pathophysiology of neonatal HI is characterized by a succession of stages (Figure 1). At the cellular level, HI induces energy failure in two steps, separated by a latency phase. The primary step is an immediate reaction to the reduction in blood flow, and therefore the supply of O_2_ and energy substrates to the brain. This process induces a drastic drop in the production of adenosine triphosphate (ATP), which leads to a failure of the sodium/potassium ATP-dependent pump (Na^+^/K^+^ATPase) [1] followed by a depolarization of the cell. In turn, this induces the release of excitatory amino acids, mainly glutamate. Massively released glutamate binds to its receptors, causing a huge influx of Ca^2+^ and Na^+^ into the cell [2].

This Ca^2+^ entry is deleterious and leads to cerebral edema and microvascular damage, resulting in apoptosis or necrosis, depending on the severity of the HI episode. Necrosis occurs in very severe conditions of neonatal HI [3]. It causes cell swelling and membrane rupture, leading to cell death. When membranes break down, the cell content is released, causing further inflammation. This inflammation is responsible for an influx of microglial cells that release pro-inflammatory factors [4]. In turn, these factors can damage the white matter and lead to the formation of scar tissue.

If HI is less severe, cells will undergo apoptosis, with preservation of the membrane and without additional inflammation [2]. Reperfusion is followed by a latency period of 1 h to 6 h, during which the impairment of oxidative metabolism can be resorbed before the irreversible failure of mitochondrial function [3]. The duration of this phase depends on the severity of the HI episode: the more severe the HI, the shorter the phase [5]. This latency phase corresponds to the therapeutic window [6]. It is during this therapeutic window that moderate hypothermia is established—only for newborns at over 35 weeks of gestation—by lowering the core temperature (either at the head level only or throughout the whole body) to 33 °C to 34 °C for 72 h, then slowly and gradually warming the newborn for 6 h to 12 h. 

After the latency phase, a second energy deficit occurs. It is mainly characterized by oxidative stress and inflammation. The exacerbated production of free radicals is responsible for oxidative stress, which causes damage to the membranes of neurons and leads to necrosis or apoptosis. This oxidative stress is particularly deleterious for a newborn’s brain [7], due to the low concentration of antioxidants at this age and the high consumption of O_2_ that is induced by the transition from fetal to neonatal life [8]. Newborns also have a high concentration of unsaturated fatty acids that are precursors of free radicals [9]. In addition, protein-bound iron is released, as Fe^2+^, and reacts with peroxides to form free radicals, increasing neuronal tissue damage [7]. 

Excitotoxicity is also characteristic of this step, due to an excess of extracellular neurotransmitters, in particular glutamate, just as during the first energy deficit. Glutamate overstimulates its receptors, inducing, for the second time, a massive influx of Ca^2+^ and Na^+^. The necrosis and apoptosis resulting from these two successive energy deficits lead to neuronal death, which is responsible for sensorimotor and cognitive deficits. 

Treatment of neonatal HI is primarily symptomatic and is essentially based on maintaining fluids, electrolytes, and respiratory homeostasis. The only interventional therapy that is currently applied is moderate therapeutic hypothermia [10]. However, 44% to 53% of neonates do not respond to such therapy [11]. Confronted by this situation, any opportunity for developing new therapeutic avenues is a priority. Among the molecules with high neuroprotective potential, resveratrol has drawn particular attention, because numerous studies have demonstrated its anti-inflammatory and antioxidant properties, as well as its capacity to modulate the intracellular effectors that are involved in neuronal cell metabolism and survival/death. Resveratrol and its derivatives are promising, as the latest studies have demonstrated a neuroprotective transgenerational and nutritional application in the context of neonatal HI. Here, we review the in vivo elements that favor a nutraceutical or therapeutic development of the use of these compounds to counteract the deleterious effects of HI. We develop very recent in vivo innovative research on the use of these molecules.

## 2. Methods

In writing this review, the PubMed database was screened with the following terms: “neonatal hypoxia-ischemia” (5305 results) and “neuroprotection” (coupled with NHI: 1478 results). To identify articles referring specifically to the topic of our review, we used the term “resveratrol” (coupled with NHI: 15 results) and the term “polyphenols” (coupled with NHI: 19 results). Among the articles identified in the last two searches, only the studies carried out in vivo were considered (n = 17).

## 3. Resveratrol and Derivatives

Resveratrol (3,5,4′-trihydroxystilbene, RSV) is a natural non-flavonoid polyphenol belonging to the stilbene family. Stilbenes are involved in plant defense responses and demonstrate protective properties against plant fungal pathogens and nematodes. RSV is formed from phenylalanine through the action of a key enzyme, the stilbene synthase, which couples 4-hydroxycoumaryl-CoA and three molecules of malonyl-CoA. The resulting RSV molecule consists of two aromatic rings that are connected by a methylenic bridge (Figure 2). It exists as two isomers: trans-RSV and cis-RSV. Trans-RSV has greater stability and biological activity. The cis-RSV form occurs due to the isomerization that follows the breakdown of the RSV molecule, which is caused by UV light or under high-pH conditions [12]. The bioavailability of RSV, which is defined as the digested, absorbed, and metabolized fraction that reaches the bloodstream and targets tissues to exert biological activity, is very low (less than 1%). Indeed, if RSV is well-absorbed in the intestine (i.e., 70% or more), it is present in a very low concentration (nanomolar to micromolar) in the bloodstream, because it is rapidly metabolized by glucuronidation and sulfation on the hydroxyl groups. [13].

Although RSV is the most studied of the polyphenols, a scientific interest in some of its derivatives, such as pterostilbene (trans-3,5-dimethoxy-4′-hydroxystilbene, PTE) and ε-viniferin (VNF), is emerging, due to their potentially better bioavailability. Surprisingly, however, few previous studies focused on the effects of piceatannol (trans-3′,4′,3,5-tetrahydroxy-stilbene, PIC), whose greater bioavailability that of RSV has been proven [14]. Structurally, PTE, a dimethylether analog of RSV, differs by the presence of two methoxy groups, instead of the hydroxyl groups. These additional methoxy groups confer to PTE better lipophilicity, absorption, cellular uptake, and bioavailability, compared with RSV [15,16]. VNF is a dehydrodimer of RSV; this difference in structure could improve its properties, compared with those of RSV, as well as reduce its metabolism [17].

The neuroprotection of RSV and some of its derivatives (Figure 2) has been shown in the context of neonatal HI [18,19,20].

## 4. Biological Activity

### 4.1. Anti-Apoptotic Properties

The anti-apoptotic properties of RSV require the activation of the silent mating-type information regulation 2 homolog (SIRT1), a nicotine adenine dinucleotide (NAD^+^)-dependent deacetylase, which would induce an inhibition of the transcription factor nuclear factor-kappa B (NF-κB) via a negative regulation of the signaling pathway involving the extracellular signal-regulated kinase 1 and 2/mitogen-activated protein kinase (Erk1/2 MAPK) complex. Through this pathway, RSV (pre-treatment and post-treatment) causes an overexpression of the anti-apoptotic gene B-cell lymphoma 2 (Bcl2) and a decrease in the expression of the pro-apoptotic gene Bcl-2–associated X (Bax), as well a decrease in the metalloproteases [18,21,22,23,24]. In addition, the activation of SIRT1, which leads to inactivation of the transcription factor p53 by deacetylation of a lysine residue (Lys382) [25], leads to an increase in the production of anti-apoptotic molecules and a decrease in the production of pro-apoptotic molecules, such as caspase 3 [24,26]. A decrease in caspase 3 expression was also measured 12 h after neonatal HI when RSV was injected (intraperitoneally) 10 min before the insult [27]. The HI-induced rise in intracellular Ca^2+^ levels is responsible for an increase in calpain activities, resulting in the degradation of spectrin, microtubule, and neurofilament proteins. Their high activity disrupts intracellular axonal transport and membrane functions that are involved in the stability of dendritic and axonal processes [28]. Their involvement in excitotoxic neuronal damage and necrosis has been widely demonstrated [29,30]. Moreover, in the context of HI, calpains would activate caspase 3. However, it has been shown, in a model of middle cerebral artery occlusion in rats, that RSV neuroprotection involves the inhibition of calpain activities [31].

### 4.2. Antioxidant Properties

During the reperfusion period after a neonatal HI insult, the abrupt reestablishment of the O_2_ supply leads to an overproduction of reactive oxygen species (ROS), which are involved in the oxidative damage of macromolecules, such as nucleic acids, proteins, and lipids, which can lead to cell death [32]. This oxidative stress and subsequent inflammatory reaction due to an HI event are the major contributors to brain damage [33,34,35]. RSV exhibits antioxidant properties that involve the regulation of several signaling pathways (Figure 3). In the context of neonatal HI, it was demonstrated that maternal supplementation with RSV (0.15 mg/kg/d) induces an increase in superoxide dismutase (SOD2) and SIRT1 [18]. These results are in line with the data of Shin et al., which showed that RSV is neuroprotective in the context of stroke via a stimulation of the SIRT1-PGC-1α (peroxisome proliferator-activated receptor-gamma coactivator 1α) signaling pathway [36]. PGC-1α is a regulator of mitochondrial biogenesis and oxidative stress [37]. Its activation leads to an antioxidant effect by the induction of SOD2 and uncoupling protein 2 (UCP2).

In another study, using P14 rats as a model of neonatal HI, the administration of RSV (20 mg/kg or 40 mg/kg for seven consecutive days prior to the induction of HI) reduced oxidative stress, in part by increasing the expression of glutathione peroxidase (GPx), SOD2, and catalase (CAT) via the nuclear factor erythroid 2 related factor 2/heme oxygenase 1 (Nrf2/HO-1) pathway just after the HI accident [38]. SOD2 allows the conversion of superoxide (O_2_^−^) to hydrogen peroxide (H_2_O_2_), while GPx and CAT catalyze the reduction of H_2_O_2_ to H_2_O [39]. The direct antioxidant effect of RSV is attributed to the ability to remove free radicals by scavenging ROS. This property is due to the presence of three hydroxyl groups in positions 3, 4, and 5, the presence of two aromatic rings, and a double bond in the molecules [40,41,42]. These data are in agreement with the decrease in ROS production, measured after injection of RSV (intraperitoneally, 20 mg/kg) 10 min before the neonatal HI event [33].

### 4.3. Anti-Inflammatory Properties

Inflammation is an important contributor to brain damage that is induced by HI. The anti-inflammatory properties of RSV involve several signaling pathways (Figure 4), including the SIRT1 pathway, which induces a negative regulation of the transcription factor NF-κB via the inhibition of the Erk1/2 MAPK complex [47]. In addition to direct allosteric activation, RSV can activate SIRT1 by inhibiting the activity of phosphodiesterase (PDE), which leads to an increase in 3′, 5′- cyclic adenosine monophosphate (cAMP) [48]. The increase in the cAMP level activates protein kinase A (PKA), which in turn phosphorylates and activates SIRT1 [46]. Moreover, the RSV-dependent activation of SIRT1 leads to anti-inflammatory effects through the negative regulation of the NOD-like receptor family, pyrin domain containing 3 (NLRP3) expression, involving inhibition of the NF-κB pathway by desacetylation of p65 Lys310 [38,49]. This inhibition of activated NLRP3 inflammasomes prevents the catalytic activation of caspase 1 and the release of the pro-inflammatory cytokines IL-1β and IL-18 [50]. In addition, RSV attenuates inflammation by inducing autophagy through the activation of the transcriptional factor, f transcription factor EB (TFEB) [51]. The anti-inflammatory effect also depends on the estrogen receptor ERα, on which RSV can bind allosterically, leading to the inhibition of the NF-κB pathway [52]. Coupled with its antioxidant properties, Nrf2 plays a pivotal role in inflammation by inducing the HO-1 gene, resulting in the inhibition of the NF-κB signaling pathway [53], and by inhibiting the production of pro-inflammatory cytokines, including IL-1β and IL-6 [38,54]. The expression of tumor necrosis factor alpha (TNFα), a pivotal actor in inflammatory processes, is also decreased by the administration of RSV (intraperitoneally, 20 mg/kg, 10 min before HI or 20 mg/kg/d to 40 mg/kg/d during the 7 days preceding the insult) [27,38].

## 5. Direct RSV Administration for Neuroprotection

Oxidative stress [55,56] and inflammation [57,58] are two critical mechanisms that are involved in neonatal HI brain injury. The neuroprotective role of RSV, especially via the activation of the SIRT1 signaling pathways, has been well described in the context of hypoxia, involving antioxidant, anti-apoptotic, and anti-inflammatory properties [36,59,60,61] (Figure 5). Due to these neuroprotective properties, RSV has generated growing interest in research in order to develop strategies to counteract HI-induced brain damage. Most of these studies evaluated the effect of intraperitoneal (i.p.) administration of RSV. Two kinds of approaches have been considered, in equal proportion: (i) a preventive approach, with an administration of RSV pre-HI, and (ii) a curative approach, with an administration of RSV post-HI.

### 5.1. Preventive Strategies

Different studies have evaluated the neuroprotective effect of RSV through a preventive strategy: e.g., RSV being administrated to pups before the HI event. All of these studies are summarized in Table 1. The administrated doses of RSV ranged from 200 µg/kg to 40 mg/kg; the most frequent dose was 20 mg/kg. For all of the considered studies, the mode of administration was an intraperitoneal (i.p.) injection and the treatment window covered a wide period ranging from 7 d to only 10 min before the HI event. Although the hypoxic levels (% O_2_) were quite similar, the duration of hypoxia varied from 45 min to 2.5 h. In all these studies, RSV administration was neuroprotective, showing a decrease in oxidative stress [33,38,62], a decrease in inflammation [27,33,38], and a limitation of neuronal death [26,27], leading to a decrease in cerebral injury [27,33,38]. Interestingly, Toader et al. combined RSV pre-treatment (during the 7 d before HI, 20 mg/kg, i.p.) with hypothermia (rectal temperature 33 °C to 34 °C for 3 h), which is the gold standard in human newborn care [62]. Data showed a synergistic neuroprotection involving a decrease in oxidative stress. Only one study evaluated this RSV neuroprotection using behavioral tests [33]. At P90, Artega et al. measured the ability of RSV administration to counteract the deleterious effects of HI on the anxiety, neophobia, and visual episodic memory of rats which had undergone an HI event at P7. All of tested cognitive functions were preserved by pre-treatment with RSV. In the same study, the efficiency of pre-treatment with RSV was compared with that of post-treatment with RSV. In the study’s conditions (RSV: 20 mg/kg, i.p. injection just after HI), the authors could not highlight any effect of RSV post-treatment. This lack of efficiency could be attributed to the dose used or the time window of administration. In fact, some other studies were able to demonstrate the neuroprotective role of a post-HI administration of RSV.

### 5.2. Curative Strategies

The curative neuroprotective effect of RSV has been tested in five different studies (Table 2). Greater experimental condition variabilities were noted in these studies, compared with the conditions used for preventive treatment. The doses used ranged from 10 mg/kg/d to 100 mg/kg/d. The time of hypoxia was from 45 min to 2.5 h. The time window of the treatment was from 0 h (just after the HI) to 14 d. While RSV was mostly administered intraperitoneally, in one study the authors used oral gavage [63]. That study also used the lowest doses of RSV, in order to mimic what would be an acceptable dose for humans. However, the similarities between the times of hypoxia and the experimental tests, in some studies, allowed for some comparisons. Pan et al. and Bian et al. used a 2.5 h duration for hypoxia and administrated 100 mg/kg of RSV (i.p., 0 h, 8 h, 18 h, 48 h, and 72 h and 0 h, 8 h, 18 h post-HI, respectively) [23,24]. Both studies investigated the modulation of Bax expression in their HI model. In these two studies, the administration of RSV after the HI event led to a decrease in Bax expression and a decrease in the associated apoptosis. In three of the studies, pups underwent a hypoxic event at 8% O_2_/92% N_2_ [64,65], or 10% O_2_/90% N_2_ (which are close values), during 1 h. In all three studies, memory was evaluated using the Morris water maze. All data showed that RSV reversed HI-induced learning and memory impairments. Moreover, even if the Morris water maze was not performed at the same age (from P28 to P33), mice that were treated with RSV during 14 d (P14 to P28; RSV: 40 mg/kg/d; oral gavage) performed as well as mice treated at 0 h, 24 h, and 48 h after HI (P7 to P9; RSV: 100 mg/kg/d; i.p. injection). Those results highlighted that a lower dose administered during a longer period can be as neuroprotective as a higher dose administered during a shorter time. Moreover, as similar neuroprotection by RSV is provided by i.p. injection and force-feeding, force-feeding should be preferred.

Altogether, these data suggest that RSV can be a promising therapeutic drug to counteract brain damage that is linked to neonatal HI injury, in both preventive and curative administration. Although these previous studies elegantly demonstrated a neuroprotective role of RSV in the context of neonatal HI, two major limitations precluded an eventual transfer to human newborns: (i) the high doses used, and (ii) the mode of administration, which was mainly by i.p. injection. In order to promote a clinical translation of RSV treatment to offer neuroprotection in a context of HI, the neuroprotective effect of RSV was tested at a very low dose (0.15 mg/kg/d) through maternal supplementation. 

## 6. Maternal Supplementation for Neuroprotection

### 6.1. Maternal Supplementation with RSV

Isac et al. were the first to test the neuroprotective effect of maternal supplementation with RSV [66]. They did not used a neonatal HI model, but rather an asphyxia model: pups were exposed to a mixture of gas composed of 9% O_2_, 20% CO_2_, and 71% N_2_, for 90 min. For maternal supplementation, female Wistar rats received an RSV-enriched diet at a dose of 50 mg/kg/d, beginning at P30. Rats were mated at an age between P90 and P100 and the enriched diet was maintained until the offspring were 7 d old. Taking into consideration the intergenerational aspect, the doses used in that study were much lower than the doses used in previous studies with oral gavage or direct i.p. injection. The RSV-enriched maternal diet led to hippocampal neuroprotection of the offspring in the context of asphyxia. The maternal RSV-enriched diet decreased neuronal inflammation in the offspring after asphyxia by decreasing IL-1β and TNFα production. RSV also decreased protein S-100B expression, an increase that can be considered as a marker of brain injury [67].

Moreover, the RSV-enriched diet prevented hippocampal damage by acting at the epigenetic level through a decrease in miR-15a expression, which led to accelerated neuronal growth and maturation. Even if the doses used in the study by Isac et al. were lower than those that were usually used, they were still relatively high.

An innovative approach was recently developed to counteract the deleterious effect of neonatal HI. The neuroprotection of a maternal supplementation with RSV was evaluated, as it was in the study by Isac et al. The innovative aspect of this approach was the nutritional consideration, with extremely low doses in the order of what would be found in a standard diet. In that study, dams were supplemented via drinking water with 0.15 mg/kg/d of RSV during different time windows, either before or after the HI event (before HI: during the last week of gestation or the first week of breastfeeding, or both; after HI: during the week after the HI event) [18]. If the curative supply of RSV was neuroprotective, the best neuroprotection was expected preventatively, over the longest period (i.e., during the last week of gestation and the first week of breastfeeding). This neuroprotection was characterized by (i) a decrease in brain lesion volume, (ii) a decrease in the severity of edema, and (iii) a preservation of the cognitive-sensori-motor abilities of the pups. The anti-apoptotic pathway via the increase in SIRT1 and Bcl2, as well as the antioxidative pathway via the overexpression of SOD, were involved, despite the low doses used. Moreover, the neuroprotective effects of the maternal supplementation involved the stimulation of brain energy metabolism in the pups (Figure 6). These results were the first to show a stimulation of cerebral energy metabolism by maternal supplementation with resveratrol, in the context of neonatal HI. Interestingly, the results can be related to the neuroprotection offered by post-HI lactate administration [68]. In particular, RSV maternal supplementation induced an increase in the expression of the main components of the astrocyte–neuron lactate shuttle (ANLS)—the monocarboxylate transporter 2, MCT2; the lactate dehydrogenases a and b, LDHa, LDHb; the glutamate–aspartate transporter, GLAST; the glutamate transporter, GLT1; and the Na^+^/K^+^ ATPase). This stimulation of the ANLS may spare glucose, benefiting the pentose phosphate pathway and the neuronal red/ox balance. These studies, which involved maternal supplementation, are summarized in Table 3.

Although accumulating preclinical studies show a neuroprotective effect of RSV in numerous pathologies—obesity, diabetes, metabolic syndrome, cancer, cardiovascular diseases, and neuronal pathologies including HI—the therapeutic application in humans remains limited due to the short half-life of RSV, its rapid metabolism (into glucuronide and sulfate or methoxy derivatives) and high urinary elimination. The dichotomy between data from preclinical studies and clinical studies has been reviewed previously [69,70]. Despite very promising results in preclinical studies following the administration of high, or even low, doses of RSV, data from clinical studies are less convincing. However, it should be emphasized that clinical studies have, for the most part, been conducted on small cohorts with different protocols, a wide range of resveratrol doses, and short periods of time. Moreover, the best results with RSV were obtained when RSV was administrated as part of polyphenol-rich extracts. These results suggested that beneficial effects may be attributed to a synergistic action between RSV and other phenolic compounds, rather than RSV alone. An increase in bioavailability would result in an increase in bio-efficacy. Two approaches have been envisaged in order to increase bio-efficacy: the use of an RSV derivative that has a higher bioavailability, or the use of a polyphenolic cocktail that exerts synergistic effects. Both strategies were tested.

### 6.2. Maternal Supplementation with PIC

To increase the bio-efficacy and neuroprotection through maternal supplementation, the effect of PIC, which is four times more bioavailable than RSV, was tested in rats [19] (Table 4). The PIC neuroprotective properties were compared with those of RSV in a more deleterious context when neonatal HI was combined with maternal alcoholism [71]. The data showed a strong neuroprotection that was obtained by maternal supplementation with PIC (0.15 mg/kg/d, in the drinking water), characterized by (i) a decrease in brain lesion volume, (ii) a decrease in the severity of edema, (iii) a long-term rearrangement of the white matter, and (iv) a preservation of the cognitive-sensory-motor abilities of the pups. The most convincing result was with respect to the incidence of brain lesions. Indeed, the incidence of brain lesions was 80% for pups whose dam had been supplemented with RSV (0.15 mg/kg/d, during the last week of gestation and the first week of breastfeeding), whereas it was 55% for pups whose dam had been supplemented with PIC (same dose, same time window for maternal supplementation). These data demonstrate that better bioavailability is linked to better bio-efficacy.

### 6.3. Maternal Supplementation with Multiple Polyphenols

Another strategy to increase bio-efficacy is to take advantage of putative synergistic effects of a combination of different polyphenols (Table 5). An assembly of polyphenols or a polyphenol associated with its food source could show stronger beneficial effects than an isolated polyphenol, in terms of complementarity and synergy. Feng et al. showed that i.p. administration of grape seed extract, which is rich in polyphenols, including RSV (5.89 μg/g dry weight [72]), was neuroprotective [73]. Injections of 50 mg/kg of grape seed extract, before or after the neonatal HI event, led to a decrease in apoptosis and brain lesions, as well as a decrease in lipid peroxidation [73,74]. However, as in most studies, polyphenols were administered by i.p. injections, which is not realistic for translation to human neonates. To increase the bio-efficacy of RSV, polyphenols extracted by green chemistry were administered at nutritional doses, via a supplementation of pregnant and/or breastfeeding rat females (RSV, 0.15 mg/kg/d; PTE, 0.15 mg/kg/d and ε-VNF, 0.30 mg/kg/d) [20]. In this study, a neuroprotective effect of maternal supplementation with eco-sustainably extracted grape polyphenols was demonstrated for the first time in the context of neonatal HI. Compared with the maternal supplementation with RSV alone, the polyphenolic cocktail induced a better striatal neuroprotection. 

Polyphenols are produced as needed by plants, but they cannot be synthesized by the human body; the intake of these micronutrients is therefore from the diet. In order to compare and determine an order of magnitude of the doses tested in the studies mentioned above, we evaluated the quantities of elements rich in resveratrol, commonly found in a diet, which should be consumed by a pregnant woman to match the amount administered in animal experiments. Quantities are given for a pregnant woman weighing 70 kg (Table 6). In contrast to the classically high doses used in animal experiments, 0.15 mg/kg/d is a nutritional dose that is realistically achievable through a natural food supply. Moreover, RSV can cross the blood–brain barrier (RSV: i.p., 30 mg/kg, [75]), as well as the placental barrier. Its consumption during pregnancy did not show any deleterious effects in animals [76], nor in pregnant women [77]. However, the non-toxic character of RSV is controversial, as some deleterious effects have been reported in a few studies in vitro and in vivo (non-pregnant condition). This duality was recently very well discussed by Shaito et al. [78]. In that review article, the authors highlighted the importance of the doses of RSV used to observe either beneficial (at low doses) or deleterious (at high doses) effects [79,80], a phenomenon called hormesis. Indeed, most of the toxic effects of RSV, in particular the pro-oxidant effect, were observed at high doses [81,82,83], indicating the use of low doses in preclinical studies in order to permit further translation to the clinic by precluding any possible toxic effect. 

## 7. Conclusions and Future Perspectives

Due to its anti-apoptotic, antioxidant, anti-inflammatory properties, and its ability to stimulate cerebral energy metabolism, RSV is a very good candidate to counteract the deleterious effects of neonatal HI. However, its low bioavailability makes the translation from preclinical studies to human neonates difficult. An increase in polyphenolic bioavailability, in correlation with an increase in bio-efficacy, would be a very interesting avenue for transposing the benefits of RSV to humans. In addition, from the perspective of clinical application, some points should be considered for future studies: (i) the doses used should be low and of the nutritional order; and (ii) the mode of intraperitoneal administration should be abandoned in favor of maternal supplementation. The experimental conditions (doses, hypoxia time, and administration window) should be standardized in order to be able to compare the different conditions among the studies. The overall molecular mechanisms remain to be elucidated. The behavioral assessments of the beneficial effects of preventive administration of polyphenols could be substantiated in order to better understand the mode of action of polyphenolic neuroprotection. More studies on the neuroprotective effects of maternal supplementation with polyphenolic cocktails or plant extracts rich in RSV-derived polyphenols could open new, promising, and innovative avenues for counteracting the deleterious effects of neonatal HI.

## Figures and Tables

**Figure 1 nutrients-14-03793-f001:**
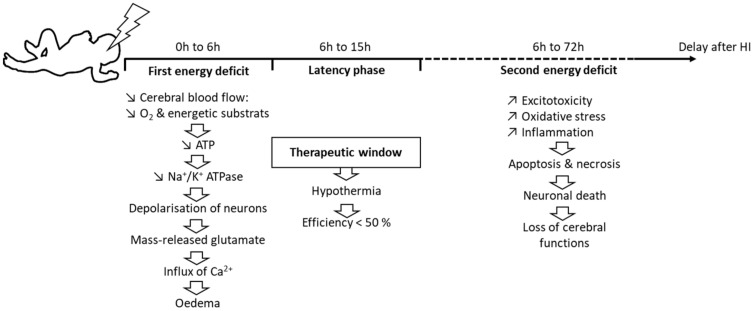
Dynamic events during neonatal HI.

**Figure 2 nutrients-14-03793-f002:**
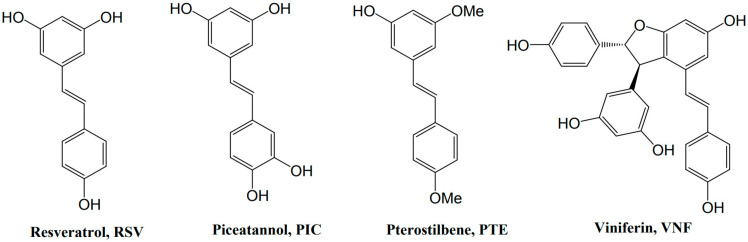
Formulas of stilbenes whose neuroprotection has been shown in HI.

**Figure 3 nutrients-14-03793-f003:**
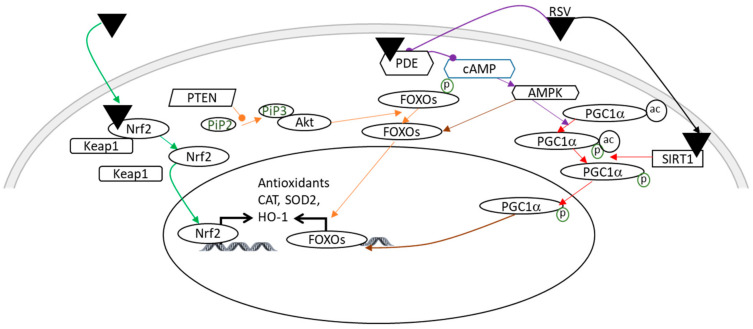
Illustration of the main antioxidant effects of RSV, which may be involved in its neuroprotection in a context of HI. A color code linking the signaling pathways to their bibliographic references is used: green arrows [38]; orange arrows [43,44]; purple arrows [45,46]; red arrows [36]; and a brown arrow [37]. Pointed arrows: stimulation; rounded arrows: inhibition. Ac, acetylation; Akt, Akt kinase; AMPK, AMP-activated protein kinase; cAMP, cyclic adenosine monophosphate; CAT, catalase; FOXO, forkhead Box O3; HO-1, heme oxygenase; Keap1, kelch-like ECH associated protein; Nrf2, nuclear factor erythroid 2 related factor 2; P, phosphorylation; PDE, phosphodiesterase; PGC1α, peroxisome proliferator-activated receptor-gamma coactivator 1α; PTEN, phosphatase and tensin homolog; RSV, resveratrol; SIRT1, silent mating type information regulation 2 homolog; SOD2, super-oxide dismutase.

**Figure 4 nutrients-14-03793-f004:**
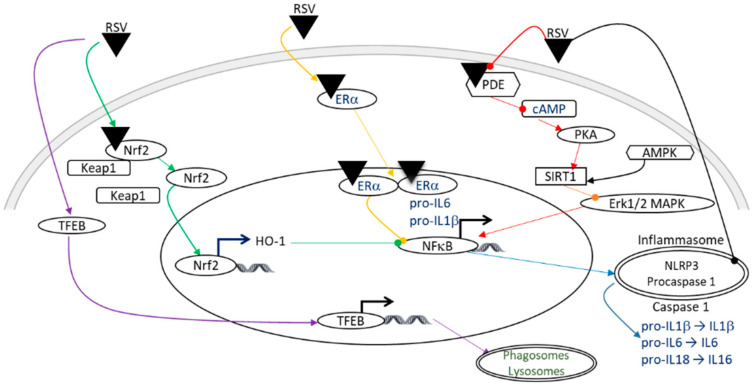
Illustration of the anti-inflammatory effects of RSV, which may be involved in neuroprotection in the context of HI. A color code linking the signaling pathways to their bibliographic references is used: purple arrows [51]; green arrows [38,53]; yellow arrows [52]; red arrows [46,48]; an orange arrow [47]; and blue arrows [38,49,50]. Pointed arrows: stimulation; rounded arrows: inhibition. AMPK, AMP-activated protein kinase; cAMP, cyclic adenosine monophosphate; ER, estrogen receptor; Erk1/2, extracellular signal-regulated kinase 1 and 2; HO-1, heme oxygenase; IL, interleukin; Keap1, kelch-like ECH associated protein; MAPK, mitogen-activated protein kinase; NF-κB, nuclear factor-kappa B; NLRP3, NOD-like receptor family, pyrin domain containing 3; Nrf2, nuclear factor erythroid 2 related factor 2; PDE, phosphodiesterase; PKA, protein kinase A; RSV, resveratrol; SIRT1, silent mating type information regulation 2 homolog; TFEB, f transcription factor EB.

**Figure 5 nutrients-14-03793-f005:**
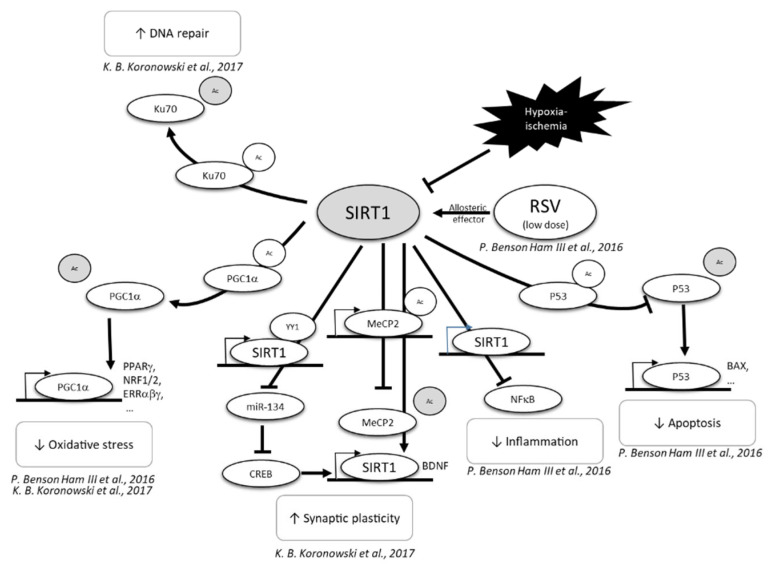
Neuroprotective effect of RSV, through SIRT1 activation [59,61]. CREB, C-AMP response element-binding protein; MECP2, methyl CpG binding protein 2; NF-κB, nuclear factor-kappa B; P53, tumor; PGC1α, peroxisome proliferator-activated receptor-gamma coactivator 1α; PPP, pentose phosphate pathway; RSV, resveratrol; SIRT1, sirtuin 1.

**Figure 6 nutrients-14-03793-f006:**
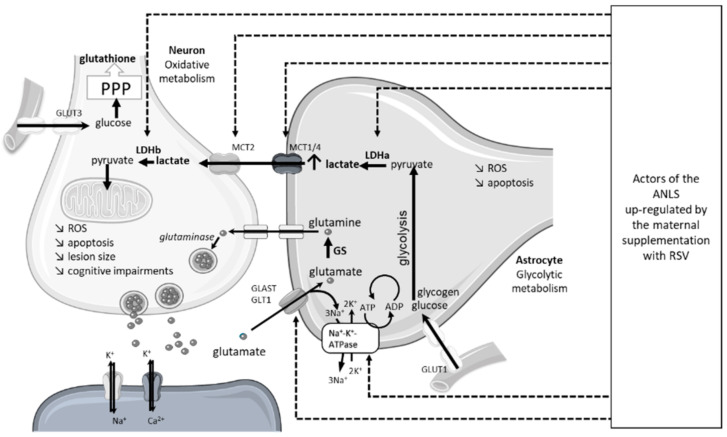
Molecular targets in offspring ANLS upregulated by maternal RSV supplementation. ANLS, astrocyte-neuron lactate shuttle; GLAST, glutamate aspartate transporter 1; GLT1, astrocytic glutamate transporter 1; GLUT, glucose transporter; GS, glutamine synthetase; LDH, lactate dehydrogenase; MCT, monocarboxylate transporter; PPP, pentose phosphate pathway; ROS, reactive oxygen species; RSV, resveratrol.

**Table 1 nutrients-14-03793-t001:** In vivo evaluation of the neuroprotective effects of pre-treatment with RSV, in the context of neonatal HI.

Polyphenols	Species	Age of HI	Ischemia	Recovery	Hypoxia	Doses	Mode of Administration	Time of Administration	Main Results	Ref.
RSV	Rat (n = 48)	P7	Common carotid artery ligation	1 h	2.5 h 8% O_2_ 92% N_2_	20 mg/kg/d 40 mg/kg/d	i.p.	During 7 d before HI	↘ Cerebral edema	Gao et al., 2018 [38]
									↘ Brain lesion ↘ Lipid peroxidation	
									↘ Inflammatory markers	
									↗ Antioxidative status	
									↗ HO-1 & Nrf2	
RSV	Rat (n = 33)	P7	Cauterized common carotid	1 h	2 h 15 min 8% O_2_ 92% N_2_	20 mg/kg	i.p.	10 min before HI	↘ Caspase 3	Revuelta et al., 2017 [27]
									↘ TNFα	
									↘ GFAP	
RSV	Rat (n = 28)	P7	Left common carotid artery ligation	2 h	2 h 15 min 8% O_2_ 92% N_2_	20 mg/kg	i.p.	10 min before HI	↘ Brain lesion	Arteaga et al., 2015 [33]
									Preservation of myelination	
									↘ Astroglyosis	
									Maintenance of mitochondrial integrity	
									↘ ROS production	
									↗ Behavioral abilities	
RSV	Rat (n = 80)	P7	Clamp of the right commun carotid artery	0 h	1.5 h 9% O_2_ 91% N_2_	20 mg/kg/d	i.p.	During the 7 days before HI	↘ Oxydative stress	Toader et al., 2013 [62]
RSV	Mouse (n = 193)	P7	Common carotid artery ligation	2 h	45 min 8% O_2_ 92% N_2_	20 mg/kg 200 µg/kg	i.p.	24 h or 10 min before HI	↘ Caspase 3 ↘ Calpain	West et al., 2007 [26]

**Table 2 nutrients-14-03793-t002:** In vivo evaluation of the neuroprotective effects of post-treatment with RSV, in the context of neonatal HI.

Polyphenols	Species	Age of HI	Ischemia	Recovery	Hypoxia	Doses	Mode of Administration	Time of Administration	Main Results	Ref.
RSV	Mouse (n = 24)	P7	Right common carotid artery ligation	2 h	1 h 8% O_2_ 92% N_2_	100 mg/kg	i.p.	0 h, 24 h, 48 h	↘ Hippocampal neuronal damage	Peng et al., 2022 [64]
									↗ Dendritic spine density	
									↗ Synaptic proteins	
									Modulation of SIRT1/NF-κB axis	
									-Improvement of cognitive & memory deficit	
									↘ cerebral edema	
									↘ brain lesion	
									↘ Lipid peroxidation	
									↘ Inflammatory markers	
									↗ Antioxidative status	
									↗ HO-1 & Nrf2	
RSV	Mouse (n = 16–24)	P7	Electro-coagulation of the right carotid artery	1 h	1 h 10% O_2_ 90% N_2_	10 mg/kg 40 mg/kg	Oralgavage	P14 During 14 days	↗ Proliferation of neuronal strem cells ↗ Neuronal differentia-tion in hippocampus	Li et al., 2020 [63]
									Improvement	
									of spatial learning and memory	
									↘ Depressive and anxiety-like mood	
RSV	Rat	P7	Left common carotid artery ligation		2.5 h 8% O_2_ 92% N_2_	100 mg/kg	i.p.	0 h, 8 h, 18 h after HI	↗ miR-96	Bian et al., 2017 [23]
									↘ Bax	
									↘ Brain lesion	
RSV	Rat (n = 67)	P7	Left common carotid artery ligation	1.5 h	2.5 h 8% O_2_ 92% N_2_	100 mg/kg	i.p.	0 h, 8 h, 18 h, 48 h, 72 h after HI	Inhibition of microglia activation	Pan et al., 2016 [24]
									Anti-apoptotic effect	
									↘ Bax, Bcl-2 and caspase 3	
RSV	Rat (n = 38)	P7	Left common carotid artery ligation	2–3 h	1 h 8% O_2_ 92% N_2_	90 mg/kg	i.p.	0 h after HI	↘ Brain lesion	Karalis et al., 2011 [65]
									Improvement of behavioral abilities	

**Table 3 nutrients-14-03793-t003:** In vivo evaluation of the neuroprotective effects of maternal supplementation with RSV, in the context of neonatal asphyxia or neonatal HI.

Polyphenols	Species	Age of HI	Ischemia	Recovery	Hypoxia	Doses	Mode of Administration	Time of Administration	Main Results	Ref.
RSV	Rat (n = 126)	P7	Left common carotid artery ligation	0.5 h	2 h 8% O_2_ 92% N_2_	0.15 mg/kg/d	Maternal supplemen-tation	Last week of gesta-tion + first week of lactation	Anti-oxydativeAnti-apoptotic Stimulation of brain energy metabolism	Dumont et al., 2021 [18]
RSV	Rat (n = 24)	P6	Ø		1.5 h 9% O_2_ 20% CO_2_ 71% N_2_	50 mg/kg/d	Maternal supplemen-tation	P30 of the dams until pus were7 d old	↘ Neuroinflam-mation	Isac et al., 2017 [66]
									↘ Neuronal injury	

**Table 4 nutrients-14-03793-t004:** In vivo evaluation of the neuroprotective effects of maternal supplementation with PIC, in the context of neonatal HI.

Polyphenols	Species	Age of HI	Ischemia	Recovery	Hypoxia	Doses	Mode of Administration	Time of Administration	Main Results	Ref.
RSV or PIC	Rat (n = 78)	P7	Left common carotid artery ligation	0.5 h	2 h 8% O_2_ 92% N_2_	0.15 mg/kg/d0.15 mg/kg/d	Maternal supplementation	Last week of gesta-tion + first week of lactation	↘ Brain lesion ↗ Sensitivo-motor & cognitive abilitiesNeuroportection of PIC better than RSV	Dumont et al., 2020 [71]
PIC	Rat (n = 52)	P7	Left common carotid artery ligation	0.5 h	2 h 8% O_2_ 92% N_2_	0.15 mg/kg/d	Maternal supplementation	Last week of gesta-tion + first week of lactation	↘ Brain lesion	Dumont et al., 2019 [19]
									↗ Sensitivo-motor & cognitive abilities	

**Table 5 nutrients-14-03793-t005:** In vivo evaluation of the neuroprotective effects of maternal supplementation with grape seed extract or a polyphenolic cocktail in the context of neonatal HI.

Polyphenols	Species	Age of HI	Ischemia	Recovery	Hypoxia	Doses	Mode of Administration	Time of Administration	Main Results	Ref.
RSV+VNF+PTE	Rat (n = 58)	P7	Left common carotid artery ligation	0.5 h	2 h 8% O_2_ 92% N_2_	0.15 mg/kg/d 0.30 mg/kg/d 0.15 mg/kg/d	Maternal supplemen-tation	Last week of gesta-tion + first week of lactation	↘ Brain lesion	Roumes et al., 2022 [20]
									Preservation of sensori-motor & cognitive function High striatal neuroprotec-tion	
Grape seed extract	Rat (n = 27)	P7	Right common carotid artery ligation	2–3 h	2.5 h 8% O_2_ 92% N_2_	50 mg/kg	i.p.	5 min to 5 h post-HI + 3 doses in 26 h post-HI	↘ Brain lesion	Feng et al., 2007 [74]
									↘ Lipid	
									peroxidation	
Grape seed extract	Rat (n = 123)	P7	Right common carotidartery ligation	2–3 h	2.5 h 8% O_2_ 92% N_2_	25 or 50 mg/kg	i.p.	5 min before HI 4 h after HI Twice daily for 1 d	↘ Brain lesion	Feng et al., 2005 [73]
									↘ Lipid	
									peroxidation	

**Table 6 nutrients-14-03793-t006:** Examples of food intake necessary for a 70 kg pregnant woman to match the doses of resveratrol used in preclinical studies.

Product	Dose Used in Preclinical Studies	Food	Daily Food Intake Necessary for a Pregnant Woman
RSV	50 mg/kg/d (maternal supplementation) [66]	Fresh grape [84,85]	1–21 kg
		Fresh peanuts [86]	12–9 kg
		Cranberry juice [84]	17,500 L
		Grape juice [87,88]	241–7778 L
		Black chocolate [89]	10 kg
		Milk chocolate [89]	35 kg
	10 mg/kg/d (oral gavage of the pups) [63]	Fresh grape	200 g–4 kg
		Fresh peanuts	3–13 kg
		Grape juice	48–1555 L
		Cramberry juice	3500 L
		Black chocolate	2 kg
		Milk chocolate	7 kg
	0.15 mg/kg/d (maternal supplementation) [18,19,20,71]	Fresh grape	3–66 g
		Fresh peanuts	35–117 g
		Grape juice	4–23 L
		Cranberry juice	53 L
		Black chocolate	30 g
		Milk chocolate	100 g

## Data Availability

Not applicable.

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
