# Peer review of "Resveratrol and Some of Its Derivatives as Promising Prophylactic Treatments for Neonatal Hypoxia-Ischemia"

_nutrients, 2022, doi:10.3390/nu14183793_

Round 1
Reviewer 1 Report
Summary,
This is a well-written review article on the pre-clinical studies of polyphenols as therapy or preventive measure for neonatal hypoxia-ischemia injury. The content is appropriate, and the conclusion is sound.
Minor points:
1. All special characters are not shown correctly. For example, Line 111. NF- B should be NF-kB.
2. Figure 4. The use of colored lines is confusing. Please simplify or define the meaning of colors in the figure legends.
Author Response
Answers to reviewer 1
This is a well-written review article on the pre-clinical studies of polyphenols as therapy or preventive measure for neonatal hypoxia-ischemia injury. The content is appropriate, and the conclusion is sound.
We would like to thank reviewer 1 for his/her positive comments.
Minor points:
- All special characters are not shown correctly. For example, Line 111. NF- B should be NF-kB.
We thank the reviewer for his/her comment. We are sorry for the inconvenience caused by the typographical problems. We carefully re-read the article and found no faulty character. This is probably a software compatibility issue. The article will be sent to the publisher in pdf format, in order to avoid this type of problem. However, several dashes were missing for NF-kB; they have been added.
- Figure 4. The use of colored lines is confusing. Please simplify or define the meaning of colors in the figure legends.
We thank the reviewer for his/her remark. In figures 3 and 4, the use of colored lines is associated with the bibliographical references: each color corresponds to a signaling pathway and to the bibliographical references of the authors who highlighted these signaling pathways. This presentation is intended to be simple and helpful for the reader. Since reviewer 1 found the color code confusing, the following clarification has been made in the legend of figures 3 and 4: "A color code linking the signaling pathways to their bibliographic references has been used:"

Reviewer 2 Report
The review “Polyphenols as a Promising Prophylactic Treatment for Neonatal Hypoxia-Ischemia” by Hélène Roumes and colleagues is an interesting article covering the use of polyphenols in a medical area, neonatal hypoxic damage, that deserves some therapy urgently.
Although by the title you expect a review on several polyphenols the truth is that the article is really centered on “Resveratrol” and although that is ok, probably the title and the methodology should indicate that more clearly.
In general, my opinion is that the article tends to be overoptimistic on the results of resveratrol in this disease, and probably we need a more objective analysis of the literature results. Translation failure has been the rule in brain hypoxia.
Authors should include the sample size of animals of each included article in the tables in order to comment on how many animals have been explored so far, it seems to be a low number to make any recommendation in the field.
Authors could also comment in the discussion the situation for resveratrol in other hypoxic conditions, such as stroke or coronary disease, any advance in those fields? It seems its use is not clinically progressing, what are the reasons?.
The search strategy of included articles should be better explained, how was the search done for resveratrol and for the other polyphenols? Not clear if the search for resveratrol and for other polyphenols in the context of neonatal hypoxia was systematic or not ¿?
Authors state that “During the reperfusion period after a neonatal HI insult, the abrupt reestablishment of O2 supply leads to an overproduction of reactive oxygen species (ROS), which are involved in oxidative damage of macromolecules such as nucleic acids, proteins and lipids, which can lead to cell death [32]. This oxidative stress and subsequent inflammatory re- action due to an HI event are the major contributors to brain damages [33-35].” However, for me is not so clear and evident that reperfusion injury exists or is the main damage related factor in such type of neonatal hypoxia ¿? And in fact, that process of reperfusion injury as described by the authors is more related/described to a thrombus occluding an artery and then being reopened naturally or by different reperfusion techniques (thrombolytics or mechanical).
In fact, many consider this type of hypoxia more similar to a global hypoxia as shown later by many of the included experimental articles.
“Maternal supplementation with PIC to increase the bio-efficacy and neuroprotection through the maternal supplementation, the effect of PIC, which is four times more bioavailable than RSV, was tested in rats [19] (Table 4). PIC neuroprotective properties were compared with those of RSV in a more deleterious contexts when neonatal HI was combined with maternal alcoholism [69]. Data showed a strong neuroprotection obtained by maternal supplementation”
Do they compare it with standard resveratrol and does standard resveratrol works in this setting ?? not fully clear for me.
Not clear how authors propose to use the maternal supplementation strategy ?? should it be given to all mothers ?? to some high-risk pregnant women? Please clarify.
Author Response
Answers to reviewer 2
The review “Polyphenols as a Promising Prophylactic Treatment for Neonatal Hypoxia-Ischemia” by Hélène Roumes and colleagues is an interesting article covering the use of polyphenols in a medical area, neonatal hypoxic damage, that deserves some therapy urgently.
Although by the title you expect a review on several polyphenols the truth is that the article is really centered on “Resveratrol” and although that is ok, probably the title and the methodology should indicate that more clearly.
We thank the reviewer for this comment. There are many studies about the neuroprotective effects of polyphenols. Many polyphenols (not only resveratrol) have been tested but in most of these studies, the neuroprotective effects were evaluated on in vitro or ex vivo models. Since our review focuses on in vivo effects in neonatal hypoxia-ischemia, we only considered studies using in vivo models. In such case, only a few polyphenols have been tested: resveratrol in many of them but also some of its derivatives such as piceatannol, viniferin and pterostilbene. To better reflect the content of our review article and to avoid any confusion, we have modified the title of our article as followed: "Resveratrol but more likely some of its derivatives as promising prophylactic treatments for neonatal hypoxia-ischemia ».
In general, my opinion is that the article tends to be overoptimistic on the results of resveratrol in this disease, and probably we need a more objective analysis of the literature results. Translation failure has been the rule in brain hypoxia.
We agree with the reviewer regarding the failure of translation to the clinic for resveratrol. This point is indicated lines 267-270: « Although these studies elegantly demonstrated a neuroprotective role of RSV in the context of neonatal HI, two major limitations preclude an eventual transfer to human newborns: (i) the high doses used and (ii) the mode of administration mainly by i.p. injection. » and lines 324-327: « Although accumulating preclinical studies show a neuroprotective effect of RSV in various neuronal pathologies, and in particular in HI, the therapeutic application in humans remains limited due to the short half-life of RSV, its rapid metabolism (into glucuronide and sulphate, or methoxy derivatives) and high urinary elimination. » In order to emphasize the ineffectiveness of RSV in clinical trials, the reviewer's comment has been taken into account and lines 324-327 (now lines 332-335) have been modified: « Although accumulating preclinical studies show a neuroprotective effect of RSV in various neuronal pathologies, and in particular in HI, translation failure has been the rule in brain hypoxia. Indeed, the therapeutic application in humans remains limited due to the short half-life of RSV, its rapid metabolism (into glucuronide and sulphate, or methoxy derivatives) and high urinary elimination. ».
This being said, we remain optimistic that derivatives of resveratrol, which do not suffer from the same disadvantages, might have better chances to succeed in the clinic, at least for the condition of neonatal HI.
Authors should include the sample size of animals of each included article in the tables in order to comment on how many animals have been explored so far, it seems to be a low number to make any recommendation in the field.
Except for the study of Bian et al. (2017), for which the number of animals is not specified, the sample size of animals has been included in the tables. As can be seen, these numbers are sufficient to draw robust conclusions.
Authors could also comment in the discussion the situation for resveratrol in other hypoxic conditions, such as stroke or coronary disease, any advance in those fields? It seems its use is not clinically progressing, what are the reasons?
We thank the reviewer for his/her suggestion. The contradiction between preclinical and clinical data have been highlighted and discussed now in the manuscript lines 324-338: “Although accumulating preclinical studies show a neuroprotective effect of RSV in numerous pathologies: obesity, diabetes, metabolic syndrome, cancer, cardiovascular diseases, neuronal pathologies including HI, the therapeutic application in humans remains limited due to the short half-life of RSV, its rapid metabolism (into glucuronide and sulphate, or methoxy derivatives) and high urinary elimination. The dichotomy between data from preclinical studies and clinical studies have been reviewed previously [69, 70]. Despite very promising results in preclinical studies following the administration of high but even low doses of RSV, data from clinical studies are less convincing. However, it should be emphasized that clinical studies have, for the most part, been conducted on small cohorts, with different protocols, a wide range of resveratrol doses and over short periods of time. Moreover, the best results with RSV were obtained when RSV was administrated as part of polyphenol-rich extracts. These results suggest that beneficial effects may be attributed to a synergistic action between RSV and other phenolic compounds, rather than RSV alone.”
The search strategy of included articles should be better explained, how was the search done for resveratrol and for the other polyphenols? Not clear if the search for resveratrol and for other polyphenols in the context of neonatal hypoxia was systematic or not?
For this review, we used the Pubmed web site. We systematically searched for all studies on the effects of polyphenols in neonatal hypoxia-ischemia. As specified in the introduction, we have only considered the studies carried out in vivo. The keywords used for the search were: neonatal hypoxia-ischemia; resveratrol; polyphenols; neuroprotection; in vivo.
Authors state that “During the reperfusion period after a neonatal HI insult, the abrupt reestablishment of O2 supply leads to an overproduction of reactive oxygen species (ROS), which are involved in oxidative damage of macromolecules such as nucleic acids, proteins and lipids, which can lead to cell death [32]. This oxidative stress and subsequent inflammatory re- action due to an HI event are the major contributors to brain damages [33-35].” However, for me is not so clear and evident that reperfusion injury exists or is the main damage related factor in such type of neonatal hypoxia ¿? And in fact, that process of reperfusion injury as described by the authors is more related/described to a thrombus occluding an artery and then being reopened naturally or by different reperfusion techniques (thrombolytics or mechanical).
In fact, many consider this type of hypoxia more similar to a global hypoxia as shown later by many of the included experimental articles.
We thank the reviewer for this comment. In most of the cited studies, the model of HI was the very well documented model of Rice-Vannucci (ligation of the right or left common carotid artery followed by a hypoxia period). In this model, neither hypoxia alone (without carotid occlusion) nor ischemia does result in brain damage (Vannucci et al. J Exp Biol 2004)). After ischemia alone, a regulation of the ischemic effect was observed due to the presence of the circle of Willis that allows compensation in blood flow between the two brain hemispheres. The authors showed that glutamate excitotoxicity as well as oxidative stress play crucial roles in cell death following the insult. ROS production occurs during the recovery in room air (from 8 % O2 to 21 % O2). Evidence of increase production of ROS have been provided as a consequence of ligation and hypoxia followed by recovery (Ekert et al., Brain Research 1997). Studies using this model showed a neuroprotective effect of antioxidants (Palmer et al., Pediatric Research 1993 and Stoke 1994). In our team, we also recently showed that neuroprotection of post-HI lactate administration involved a decrease in ROS production in the cortex and the striatum (Roumes et al., JCBFM 2020). Moreover, evidence that cerebral oxidative stress is important in babies undergoing reoxygenation have been provided (Vento et al. 2003 J Pediatr 142:240-246).
“Maternal supplementation with PIC to increase the bio-efficacy and neuroprotection through the maternal supplementation, the effect of PIC, which is four times more bioavailable than RSV, was tested in rats [19] (Table 4). PIC neuroprotective properties were compared with those of RSV in a more deleterious contexts when neonatal HI was combined with maternal alcoholism [69]. Data showed a strong neuroprotection obtained by maternal supplementation”
Do they compare it with standard resveratrol and does standard resveratrol works in this setting ?? not fully clear for me.
In the study of Dumont et al. (2020), the neuroprotective effect of both RSV and PIC maternal supplementation were evaluated in a context of neonatal hypoxia-ischemia coupled with maternal alcoholism. Even if resveratrol maternal supplementation exhibited some neuroprotection, the neuroprotective effects of piceatannol supplementation were better.
Not clear how authors propose to use the maternal supplementation strategy ?? should it be given to all mothers ?? to some high-risk pregnant women? Please clarify.
We thank the reviewer for his/her question about the future potential beneficiaries of maternal supplementation. Data from early clinical evaluation make it possible to identify women with a high risk of neonatal hypoxia-ischemia (gestational diabetes, arterial hypertension, placenta previa, etc.). These women could be offered such a nutritional prophylaxis in priority. However, this recommendation is of the domain of medical practice. This review article does not pretend to provide medical indications but rather to highlight the directions that could be taken in future preclinical studies to identify the best strategy in order to facilitate an eventual translation of polyphenolic neuroprotection to humans in the context of neonatal hypoxia-ischemia.

Round 2
Reviewer 2 Report
The authors responses are appropriate and imporve undersatnding on sections of the manuscript. Authors state that they do the search using pubmed with the more relvant terms fro the review. Formally speaking this is not a methodologically strict "systematic review" so my advide is just stating that this is a "review".
Author Response
The authors responses are appropriate and imporve undersatnding on sections of the manuscript. Authors state that they do the search using pubmed with the more relvant terms fro the review. Formally speaking this is not a methodologically strict "systematic review" so my advide is just stating that this is a "review".
We are pleased to read that our reponses allow clarification, we thank both referees for helping us to improve our review. Concerning the methodology to perform the search on PubMed, we agree with the comment and carefully checked thoughout the manuscript that this term "systematic" review was not present.
